# Episodic Reinforcement Learning with Associative Memory

**Guangxiang Zhu**[1*], **Zichuan Lin**[2*], **Guangwen Yang**[2], **Chongjie Zhang**[1]
[1]Institute for Interdisciplinary Information Sciences, Tsinghua University, Beijing, China
[2]Department of Computer Science and Technology, Tsinghua University, Beijing, China
`guangxiangzhu@outlook.com`, `linzc16@mails.tsinghua.edu.cn`,
`ygw@tsinghua.edu.cn`, `chongjie@tsinghua.edu.cn`

## Abstract

Sample efficiency has been one of the major challenges for deep reinforcement learning. Non-parametric episodic control has been proposed to speed up parametric reinforcement learning by rapidly latching on previously successful policies. However, previous work on episodic reinforcement learning neglects the relationship between states and only stored the experiences as unrelated items. To improve sample efficiency of reinforcement learning, we propose a novel framework, called *Episodic Reinforcement Learning with Associative Memory* (ER-LAM), which associates related experience trajectories to enable reasoning effective strategies. We build a graph on top of states in memory based on state transitions and develop a reverse-trajectory propagation strategy to allow rapid value propagation through the graph. We use the non-parametric associative memory as early guidance for a parametric reinforcement learning model. Results on the navigation domain and Atari games show our framework achieves significantly higher sample efficiency than state-of-the-art episodic reinforcement learning models.

## 1 Introduction

Deep reinforcement learning (RL) has achieved remarkable performance on extensive complex domains (Mnih et al., 2015; Lillicrap et al., 2016; Silver et al., 2016; Schulman et al., 2017). Deep RL research largely focuses on parametric methods, which usually depend on a parametrized value function. The model-free approaches are quite sample inefficient and require several orders of magnitude more training samples than a human. This is because gradient-based updates are incremental and slow and have global impacts on parameters, leading to catastrophic inference issues.

Recently, episodic reinforcement learning has attracted much attention for improving sample efficiency of deep reinforcement learning, such as model-free episodic control (MFEC) (Blundell et al., 2016), neural episodic control (NEC) (Pritzel et al., 2017), ephemeral value adjustments (EVA) (Hansen et al., 2018), and episodic memory deep q-networks (EMDQN) (Lin et al., 2018). Episodic control is inspired by the psychobiological and cognitive studies of human memory (Sutherland & Rudy, 1989; Marr et al., 1991; Lengyel & Dayan, 2008; Botvinick et al., 2019) and follows the idea of instance-based decision theory (Gilboa & Schmeidler, 1995). It builds a non-parametric episodic memory to store past good experiences and thus can rapidly latch onto successful policies when encountering states similar to past experiences.

However, most of the current breakthroughs have focused on episodic memory and leave the association of memory largely unstudied. Previous work usually uses a tabular-like memory, and experiences are stored as unrelated items. Studies in psychology and cognitive neuroscience (Kohonen, 2012; Anderson & Bower, 2014) discover that associative memory in the hippocampus plays a vital role in human activities, which associates past experiences by remembering the relationships between them. Inspired by this, we propose a novel associative memory based reinforcement learning framework to improve the sample-efficiency of reinforcement learning, called *Episodic Reinforcement Learning with Associative Memory* (ERLAM), which associates related experience trajectories

---

*Equal Contribution

to enable reasoning effective strategies. We store the best historical values for memorized states like episodic memory, and maintain a graph on top of these states based on state transitions at the same time. Then we develop an efficient reverse-trajectory propagation strategy to allow the values of new experiences to propagate to all memory items through the graph rapidly. Finally, we use the fast-adjusted non-parametric high values in associative memory as early guidance for a parametric RL agent so that it can rapidly latch on states that previously yield high returns instead of waiting for many slow gradient updates.

To illustrate the superiority of the associative memory in reinforcement learning, consider a robot exploring in a maze to seek out the apple (place G), as shown in Figure 1. It collects two trajectory experiences starting from place A and B (i.e., blue dash line A-D-C and B-D-G), respectively. All the states of trajectory A-D-C receive no reward because the agent terminates at a non-reward state (place C). While in trajectory B-D-G, the final non-zero reward of catching an apple (place G) back-propagates to all the states of this trajectory. Episodic memory keeps a higher value of two trajectories at the intersection (place D) when taking actions toward the lower-right corner, but the other states in trajectory A-D are still 0. If an episodic memory based robot starts from place A again, it will wander around A because there are no positive values indicating the way to the goal. Thus based on the episodic memory, the robot may eventually take a policy like the green line (A-B-D-G) after multiple attempts. However, if the robot adopts associative memory, the high value in the place D collected from trajectory B-D-G will be further propagated to the start point A, and thus the robot can correctly take the red-line policy (A-D-G).

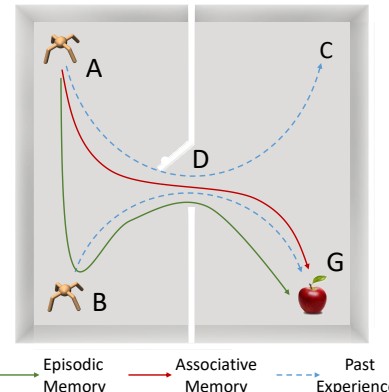

Figure 1: Comparison of selected policies based on episodic memory and associative memory. An agent starts from two places A and B, to collect two experiences.

To some extent, our associative memory is equivalent to automatic augmentation of counterfactual combinatorial trajectories in memory. Thus, our framework significantly improves the sample-efficiency of reinforcement learning. Comparisons with state-of-the-art episodic reinforcement learning methods show that ERLAM is substantially more sample efficient for general settings of reinforcement learning. In addition, our associative memory can be used as a plug-and-play module and is complementary to other reinforcement learning models, which opens the avenue for further research on associative memory based reinforcement learning.

## 2 BACKGROUND

In the framework of reinforcement learning (Sutton & Barto, 1998), an agent learns a policy to maximize its cumulative rewards by exploring in a Markov Decision Processes (MDP) environment. An MDP is defined by a tuple $(\mathcal{S}, \mathcal{A}, P, \mathcal{R}, \gamma)$, where $\mathcal{S}$ is a finite set of states, $\mathcal{A}$ is a finite set of actions available to the agent, $P : \mathcal{S} \times \mathcal{A} \times \mathcal{S} \rightarrow \mathbb{R}$ defines the transition probability distribution, $\mathcal{R}$ is the reward function, and $\gamma \in (0, 1]$ is the discount factor. At each time step $t$, the agent observes state $s_t \in \mathcal{S}$, selects an action $a_t \in \mathcal{A}$ according to its policy $\pi : \mathcal{S} \rightarrow \mathcal{A}$, and receives a scalar reward $r_t$. In the setting of finite horizon, the accumulated discounted return is calculated as, $R_t = \sum_{k=0}^{T} \gamma^k r_{t+k}$ where $T$ is the episode length and goal of the agent is to maximize the expected return for each state $s_t$.

The state-action value function $Q^\pi(s, a) = \mathbb{E}[R_t | s_t = s, a]$ is the expected return for executing action $a$ on state $s$ and following policy $\pi$ afterwards. DQN (Mnih et al., 2015) parameterizes this action-value function by deep neural networks $Q_\theta(s, a)$ and use Q-learning (Watkins & Dayan, 1992) to learn it to rank which action at is best to take in each state $s_t$ at time step $t$. The parameters of the value network $\theta$ are optimized by minimizing the $L_2$ difference between the networks output $Q_\theta(s, a)$ and the Q-learning target $y_t = r_t + \gamma \max_a Q_{\hat{\theta}}(s_{t+1}, a_t)$, where $\hat{\theta}$ are parameters of a target network that is a older version of the value network and updated periodically. DQN uses an off-policy learning strategy, which samples $(s_t, a_t, r_t, s_{t+1})$ tuple from a replay buffer for training.

DQN, as a typical parametric reinforcement learning method, suffers from sample inefficiency because of slow gradient-based updates. Thus episodic reinforcement learning is proposed to speed up the learning process by a non-parametric episodic memory. Episodic reinforcement learning enables fast learning by modeling hippocampal instance-based learning. The key idea is to store good past experiences in a tabular-based non-parametric memory and rapidly latch onto past successful policies when encountering similar states instead of waiting for many steps of optimization.

## 3    RELATED WORK

**Deep Reinforcement Learning**    Our method is closely related to DQN (Mnih et al., 2015). As the seminal work of deep reinforcement learning, DQN learns a deep neural network for state-action value function by gradient back-propagation and conducts parametric control. Following this line, a large number of extensions have been proposed to improve the learning efficiency of the parametric model. Double DQN (Van Hasselt et al., 2016) alleviates the over-estimation issue of Q-Network. Dueling network (Wang et al., 2015) separates Q-Network into two streams which predict state value and advantage value respectively and achieves better generalization across actions. Prioritized experience replay (Schaul et al., 2015b) changes the sampling priority of each training sample according to its learning error. Apart from these prior improvements, many algorithms have been proposed to accelerate reward propagation and backup mechanism. Optimality Tightening method(He et al., 2016) combines the strength of DQN with a constrained optimization approach to rapidly propagate close-by rewards. $Q^*(\lambda)$ (Harutyunyan et al., 2016) and Retrace($\lambda$) (Munos et al., 2016) incorporate on-policy samples into off-policy learning targets. Noisy Net (Fortunato et al., 2017) adds noise to the parametric model during learning to improve the exploration ability. Distributional RL (Bellemare et al., 2017) learns the value function as a full distribution instead of a expected value. Unlike these works, we focus on combining non-parametric memory and parametric model in this paper. Thus our method is complementary to these prior extensions and can be combined with them seamlessly.

**Episodic Reinforcement Learning**    Our work is also related to episodic reinforcement learning. Model-free episodic control (Blundell et al., 2016) uses a completely non-parametric model that keeps the best Q values of states in a tabular-based memory and replays the sequence of actions that so far yielded the highest return from a given start state. At the end of each episode, the Q values in memory are updated by the greater of the existing values and the accumulated discounted returns in the current episode. In the execution stage, the agent selects actions according to a k-nearest-neighbors lookup in the memory table. Recently, several extensions have been proposed to integrate episodic control with parametric DQN. Neural episodic control  (Pritzel et al., 2017) develops end-to-end episodic control by a differentiable neural dictionary to generate semi-tabular representation as slow-changing keys and then retrieves fast-updating values by context-based lookup for action selection. To better leverage the trajectory nature of experience, ephemeral value adjustments method (Hansen et al., 2018) proposes to further leverage trajectory information from replay buffer to propagate value through time and produce trajectory-centric value estimates. Our method differs from EVA in that we associate memory by a graph, and thus we can leverage not only intra-episode but also inter-episode information. Episodic memory deep q-networks  (Lin et al., 2018) distills the information of episodic memory into a parametric model by adding a regularization term in the objective function and significantly boosts up the performance of DQN. Unlike these prior works, which adopt either tabular memory or semi-tabular memory, our work builds a graph on memory items based on their relationship to form an associative memory.

**Graph Based Methods in Deep Reinforcement Learning**    Recently, several works have also been proposed to use graph for planning in deep reinforcement learning. Eysenbach et al. (2019) builds a directed graph directly on top of states in replay buffer and runs graph search to find the sequence of waypoints, leading to many easier sub-tasks and thus improve learning efficiency. Huang et al. (2019) abstracts state space as a small-scale map which allows it to run high-level planning using a pairwise shortest path algorithm. Unlike these prior works that use graphs for planning, our method reorganizes episodic memory by a graph to allow faster reward propagation. In addition, these graph-based models rely on goal-conditioned RL (Kaelbling, 1993; Schaul et al., 2015a) and only demonstrate their performance in navigation-like problems, while our approach is intended for general RL settings.

**Exploration** Efficient exploration is a long-standing problem in reinforcement learning. Prior works have proposed guiding exploration based on criteria such as intrinsic motivation (Stadie et al., 2015), state-visitation counts (Tang et al., 2017), Thompson sampling and bootstrapped models (Chapelle & Li, 2011; Osband et al., 2016), optimism in the face of uncertainty (Kearns & Singh, 2002), parameter-space exploration (Plappert et al., 2017; Fortunato et al., 2017). Recently, Oh et al. (2018) proposed self-imitation learning (SIL) and found that exploiting past good experiences can indirectly drive deep exploration. In their work, the agent imitates its own decisions in the past only when such decisions resulted in larger returns than expected. Like SIL, EMDQN (Lin et al., 2018) learns from episodic memory to replay past best decisions, therefore incentivizing exploration. In our method, we build associative memory through a graph, which enhances the exploitation of past good experiences and thus can indirectly encourage deeper exploration than EMDQN (Lin et al., 2018).

## 4 EPISODIC REINFORCEMENT LEARNING WITH ASSOCIATIVE MEMORY

### 4.1 ASSOCIATING EPISODIC MEMORY AS A GRAPH

Similar with previous episodic reinforcement learning, we adopt an episodic memory to maintain the historically highest values $Q_{\text{EC}}(\phi(s), a)$ of each state-action pair, where $\phi$ is an embedding function and can be implemented as a random projection or variational auto-encoders (VAE) (Kingma & Welling, 2013). When receiving a new state, the agent will look up in the memory and update the values of states according to the following equation,

$$Q_{\text{EC}}(\phi(s_t), a_t) \leftarrow \begin{cases} \max(Q_{\text{EC}}(\phi(s_t), a_t), R_t) & , \quad \text{if}(\phi(s_t), a_t) \in Q_{\text{EC}} \\ R_t & , \quad \text{otherwise.} \end{cases} \tag{1}$$

However, episodic memory stores states as unrelated items and does not make use of the relationship between these items. To fully exploit information in episodic memory, we further build a directed graph $\mathcal{G}$ on top of items in the episodic memory to form an associative memory, as shown in Figure 2. In this graph, each node corresponds to a memory item that records the embedded vector of a state $\phi(s)$, and we leverage transitions of states to bridge the nodes. The graph is defined as,

$$\mathcal{G} = (V, E), \ V = \phi(s), E = \{s \to s' \mid (s, a, s') \text{ is stored in memory}\}. \tag{2}$$

Given a sampled trajectory, we temporarily add each state to the graph. We add directed edges from the given state to every other previously memorized state that is the successor of it under a certain action. Our associative memory reorganizes the episodic memory and connects these fragmented states that previously yielded high returns by a graph. We rewrite these stored values $Q_{\text{EC}}(\phi(s), a)$ as $Q_{\mathcal{G}}(\phi(s), a)$ in our graph augmented episodic memory. In addition, we adopt a strategy of discarding the least recently used items when the memory is full.

### 4.2 PROPAGATING VALUES THROUGH ASSOCIATIVE MEMORY

Typical deep RL algorithms sample experience tuples uniformly from the replay buffer to update value function. However, the way of sampling tuples neglects the trajectory nature of an agent's experience (i.e., one tuple occurs after another, and thus information of the following state should be quickly propagated into the current state). EVA (Hansen et al., 2018) encourages faster value propagation by introducing trajectory-centric planning (TCP) algorithm. Nonetheless, EVA only propagates value through the current episode, which we refer to as *intra-episode* propagation. Our insight here is that one state might appear in different trajectories, and such join points can help connect different trajectories. Therefore, we explicitly build the graph between states from different trajectories in memory and thus allows *inter-episode* value propagation.

Since the graph over states is complicated (e.g., not a tree structure), value propagation over such a graph is always slow. To accelerate the propagating process, we propagate values using the sequential property. The pseudo-code of value propagation is shown in Algorithm 1. Our general idea is to update the values of the graph in the reverse order of each trajectory. Specifically, when adding a new state to the memory, we record the sequential step ID $t$ of the state at the current trajectory. For memory associating, we first sort the elements in memory by their sequential step IDs in descending order and propagate the value from states with large sequential step ID to a small one for several

**Algorithm 1** Value propagation in Associative Memory

---

$h$: embedded vector of state, $h = \phi(s)$
$\mathcal{G} \leftarrow$ Sort nodes in graph $\mathcal{G}$ by sequential step ID $t$ in descending order
**repeat**
    **for** $m = 1 \ldots |\mathcal{G}|$ **do**
        Get current state-action pair $(s, a) = (s_m, a_m)$
        Get successor state embedding $s'$ and action $a'$ using graph $\mathcal{G}$.
        Update graph augmented memory using Eq. 3.
    **end for**
**until** $Q_{\mathcal{G}}$ converges

---

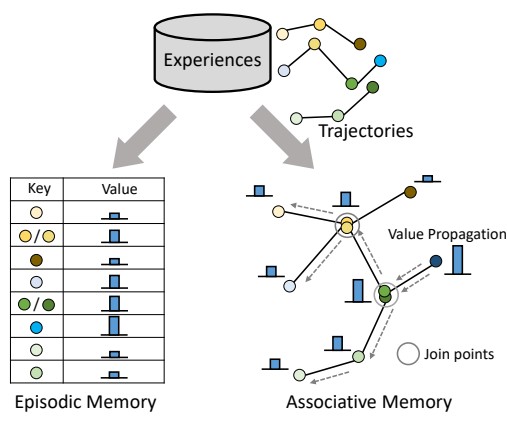

Figure 2: Comparison of episodic memory and associative memory.

iterations until $Q_{\mathcal{G}}$ values converge. At each update, we get all successor state-action pairs $(s', a')$ of the current one $(s, a)$ and current reward $r$ according to the graph $\mathcal{G}$ and apply max operation on successor action $a'$ to propagate the values to current state-action pair. Formally, our graph augmented memory is updated as follow:

$$Q_{\mathcal{G}}(\phi(s), a) \leftarrow r + \gamma \max_{a'} Q_{\mathcal{G}}(\phi(s'), a'). \tag{3}$$

Since most of states at the beginning are similar across different episodes, our reverse order updating strategy can efficiently propagate all the values of the graph. In addition, as we show in Theorem 1, our graph-based value propagation algorithm can converge to a unique optimal point. The proofs are shown in Appendix A.

**Theorem 1.** *Denote the Bellman backup operator in Equation 3 as $\mathcal{B} : \mathbb{R}^{|\mathcal{S}| \times |\mathcal{A}|} \to \mathbb{R}^{|\mathcal{S}| \times |\mathcal{A}|}$ and a mapping $Q^0 : \mathcal{S} \times \mathcal{A} \to \mathbb{R}^{|\mathcal{S}| \times |\mathcal{A}|}$ with $|\mathcal{S}| < \infty$ and $|\mathcal{A}| < \infty$, and define $Q^{k+1} = \mathcal{B}Q^k$. Repeated application of the operator $\mathcal{B}$ for our graph-based state-action value estimate $\hat{Q}_{\mathcal{G}}$ converges to a unique optimal value $Q_{\mathcal{G}}^*$.*

In the previous episodic reinforcement learning with no graph built, only the values of exactly the same or similar states can be updated. This is because in the typical update rule of episodic memory, as shown in Eq. 1, the relationship between states has been neglected. Episodic memory does not leverage the information of edges $E$ in our graph $\mathcal{G}$. Consequently, stored values in episodic memory often violate Bellman's equation. On the contrary, our associative memory allows efficient value propagation through the edges of the graph to compute the more accurate values for each state.

### 4.3 LEARNING WITH ASSOCIATIVE MEMORY

Building associative memory can be viewed as a way of augmenting counterfactual experiences. As shown in Figure 2, the same states might appear in $N > 1$ trajectories. Vanilla episodic memory maps such states to the highest values among $N$ trajectories, while our associative memory regards such states as join points to connect different trajectories, leading to totally $N^2$ trajectories. This is equivalent to sample more combinatorial trajectories from environments and thus can significantly improve sample efficiency of RL algorithms.

Our associative memory can be applied to both the learning and control phases. In this paper, we use our associative memory as guidance for the learning of the Q function. The overall framework is shown as Figure 3. Specifically, we use associative memory as a regularization term of objective function to supervise the learning of the Q network. The Q network is learned by minimizing the following objective function:

$$L_\theta = \mathbb{E}_{(s,a,s',r) \sim \mathcal{D}} \left[ \left( r + \gamma \max_a Q_{\hat{\theta}}(s', a) - Q_\theta(s, a) \right)^2 + \lambda \left( Q_{\mathcal{G}}(\phi(s), a) - Q_\theta(s, a) \right)^2 \right], \tag{4}$$

where $\lambda$ is the weight of the regularization term, $\theta$ represents parameters of parametric Q-network. Similar with DQN (Mnih et al., 2015), we also adopt a target network parameterized by $\hat{\theta}$ to stabi-

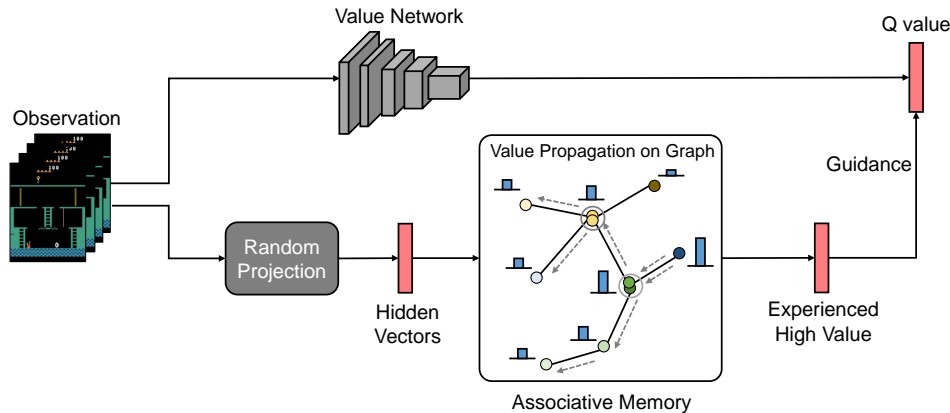

Figure 3: Overall framework of ERLAM.

lize the learning process. Through the combination of parametric and non-parametric term, we can efficiently guide the learning of a conventional Q-network by the fast-adjusted high values in associative memory so that the agent can rapidly latch on strategies that previously yield high returns instead of waiting for many steps of slow gradient update. The pseudo code of our method is shown in Algorithm 2.

---

**Algorithm 2** ERLAM: Episodic Reinforcement Learning with Associative Memory

---

$\mathcal{D}$: Replay buffer
$\mathcal{G}$: Graph (Associative memory)
$T_e$: Trajectory length of $e$-th episode
$K$: Associate frequency
**for** Episode number $e = 1 \dots E$ **do**
  **for** $t = 1 \dots T_e$ **do**
    Receive initial observation $s_t$ from environment with state embedding $h_t = \phi(s_t)$
    $a_t \leftarrow \epsilon$-greedy policy baesd on $Q_\theta(s_t, a)$
    Take action $a_t$, receive reward $r_t$ and next state $s_{t+1}$
    Append $(s_t, a_t, r_t, s_{t+1})$ to $\mathcal{D}$
    **if** $t$ mod $update\_freq == 0$ **then**
      Sample training experiences $(s, a, r, t)$ from $\mathcal{D}$
      Retrieve $Q_\mathcal{G}(\phi(s), a)$ from associative memory
      Update parameter $\theta$ using Eq. 4
    **end if**
  **end for**
  **for** $t = T_e \dots 1$ **do**
    $R_t \leftarrow r_t + \gamma R_{t+1}$, if $t < T_e$ ; $R_t \leftarrow r_t$, if $t = T_e$
    Append $(h_t, a_t, r_t, t, R_t)$ to $\mathcal{G}$ if $(h_t, a_t) \notin \mathcal{G}$
    Update $Q_\mathcal{G}$ using Eq.1 if $(h_t, a_t) \in \mathcal{G}$
  **end for**
  **if** $e$ mod $K == 0$ **then**
    Run Algorithm 1 to update $Q_\mathcal{G}$
  **end if**
**end for**

---

## 4.4 CONNECTION TO GRAPH-BASED DEEP REINFORCEMENT LEARNING

When the general RL setting used in our approach degenerates to a setting of navigation-like task that is usually adopted by goal-conditional RL (Kaelbling, 1993; Schaul et al., 2015a), the update target of associative memory in Eq. 3, $y = r + \gamma \max_{a'} Q_\mathcal{G}(\phi(s'), a')$ can be rewritten as,

$$y = \begin{cases} r & , \quad \text{if } s' \text{ is a terminal state,} \\ \gamma \max_{a'} Q_\mathcal{G}(\phi(s'), a') & , \quad \text{otherwise.} \end{cases} \tag{5}$$

Optimizing with the target in Eq. 5 is equivalent to finding the shortest path in the graph of all states. In this case, algorithm 1 is analogous to Bellman-Ford algorithm (Bellman, 1958), which is proved

that the value can converge in limited iterations. In the context of goal-conditional RL, some graph-based methods (Huang et al., 2019; Eysenbach et al., 2019) also calculated shortest path. They focus on a graph of waypoints learned by goal-conditional RL instead of memorized states that previously yield high returns. In addition, they use a parametric approach for value approximation, while we develop a non-parametric approach to improve sample efficiency of a parametric RL agent.

## 5 EXPERIMENTS

### 5.1 EXPERIMENT SETTING

We follow the same setting for network architecture and all hyper-parameters as DQN (Mnih et al., 2015). The raw images are resized to an $84 \times 84$ grayscale image $s_t$, and 4 consecutive frames are stacked into one state. The Q value network alternates convolutions and ReLUs followed by a 512-unit fully connected layer and an output layer whose size is equal to the number of actions in each game. Denote Conv($W$, $F$, $S$) as the convolutional layer with the number of filters $W$, kernel size $F$, and stride $S$. The 3 convolutional layers can be indicated as Conv(32,8,4), Conv(64,4,2), and Conv(64,3,1). We used the RMSProp algorithm (Tieleman & Hinton, 2012) with learning rate $\alpha = 0.00025$ for gradient descent training. The discount factor $\gamma$ is set to 0.99 for all games. We use annealing $\epsilon$-greedy policies from 1.0 to 0.1 in the training stage while fixing $\epsilon = 0.05$ during evaluation.

For hyper-parameters of associative memory, we set the value of $\lambda$ as 0.1 and associate frequency $K$ as 10 in the navigation domain, *Monster Kong*. In Atari games, we use the same settings for all games. The value of $\lambda$ is 0.3, and the associate frequency $K$ is 50. The memory size is set as 1 million. We use random projection technique and project the states into vectors with the dimension of $d = 4$. For efficient table lookup, we build a kd-tree for these low-dimension vectors.

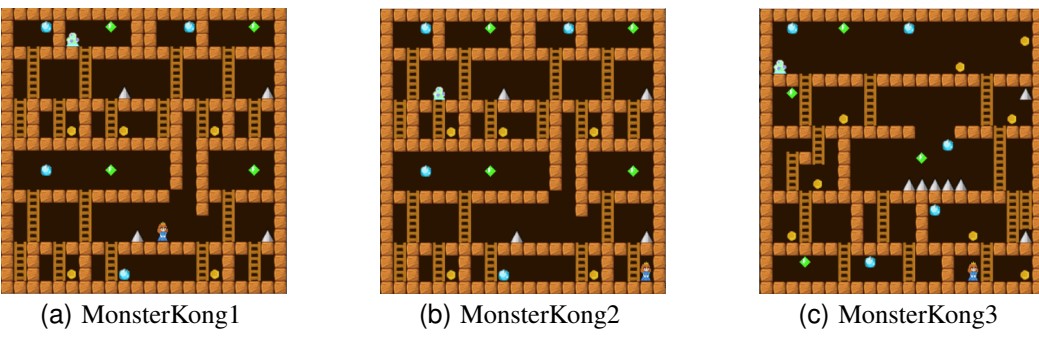

|        (a) MonsterKong1        |        (b) MonsterKong2        |        (c) MonsterKong3        |

Figure 4: Maps of *Monster Kong*. Compared to MonsterKong1, MonsterKong2 has a different goal and MonsterKong3 has a totally different MDP.

### 5.2 RESULTS ON NAVIGATION DOMAIN

We first test our model on the navigation domain, which contributes to demonstrate the superiority of our algorithm and understand the contribution of associative memory. We use a video game *Monster Kong* from Pygame Learning Environment (PLE)(Tasfi, 2016) to set up the navigation experiments. In this game, the goal of the agent is to approach the princess with actions *up*, *down*, *left*, *right*, *jump* and *noop* from random starting positions. The agent will win with an extra reward +1 when touching the princess and lose when hitting the thorns (silver triangles). We run ERLAM on three maps of *Monster Kong* (see Figure 4) and compare it with EMDQN and DQN.

As shown in Figure 5, the sample-efficiency of ERLAM significantly outperforms EMDQN and DQN. ERLAM with only 10M samples can gain higher scores than EMDQN with 80M samples on map MonsterKong2 and MonsterKong3. Then, we inspect the value estimation of Q networks and the stored values in memory to provide insight into our reinforcement learning results. We plot the average values of states in associative memory (orange line in the bottom row of Figure 5) during the training process of ERLAM. To better understand the contribution of the value propagation

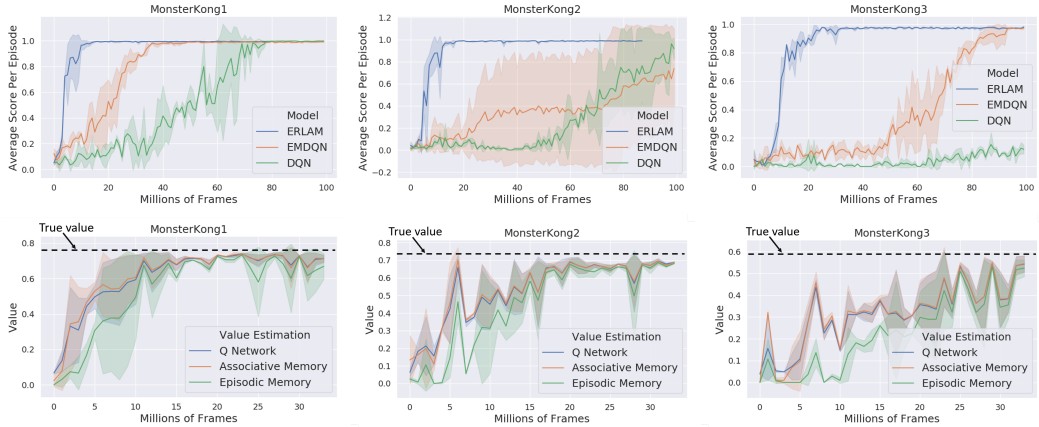

Figure 5: Learning curves of ERLAM, EMDQN, and DQN on *Monster Kong*. The top row compares the average scores per episode between all models. The bottom row shows state-action value estimates by associative memory, episodic memory, and Q networks when running ERLAM. The black dash line represents the actual discounted state-action values of the best learned policy.

process in associative memory, we maintain a memory without value propagation (which amounts to episodic memory, shown as the green line in the bottom row of Figure 5) in the meanwhile and compare the state-action values of it to associative memory. As expected, the values after value propagation of associative memory grow higher, indicating associative memory provides a better non-parametric lower bound of Q value than episodic memory. Values estimated by associative memory are closer to the true values of optimal policy (black dash line) and capable of guiding the learning of the Q network (blue line). We further visualize and compare the execution policies according to associative memory and episodic memory to gain a deeper understanding of their connections. We study a case in Figure 6. We observe that the policy provided by associative memory (yellow dash line) is exactly the combination of two policies in episodic memory (blue line and red line), and such a combinatorial trajectory is not a real trajectory in replay buffer. This result suggests that the value propagation in associative memory enables automatic augmentation of counterfactual combinatorial trajectories, which accounts for the improvement of sample efficiency in ERLAM.

## 5.3 RESULTS ON ATARI GAMES

To further evaluate the sample efficiency of ERLAM on a diverse set of games, we conduct experiments on the benchmark suite of Atari games from the Arcade Learning Environment (ALE) (Bellemare et al., 2013), which offer various scenes to test RL algorithms over different settings. We largely follow the training and evaluation protocol as (Mnih et al., 2015). We train our agents for 10 epochs, each containing 1 million frames, thus 10 million frames in total. For each game, we evaluate our agent at the end of every epoch for 0.5 million frames, with each episode up to 18000 frames, and start the game with up to 30 no-op actions to provide random starting positions for the agent.

In our experiments, we compare ERLAM with episodic reinforcement learning baselines, MFEC (Blundell et al., 2016), NEC (Pritzel et al., 2017), EMDQN (Lin et al., 2018), EVA (Hansen et al., 2018), as well as an ablation (i.e., DQN with no associative memory). MFEC directly uses the non-parametric episodic memory for action selection, while NEC, EMDQN, and EVA combine non-parametric episodic memory and a parametric Q-network. Different from previous work, ER-LAM adopts associative memory to guide the learning of a Q-network.

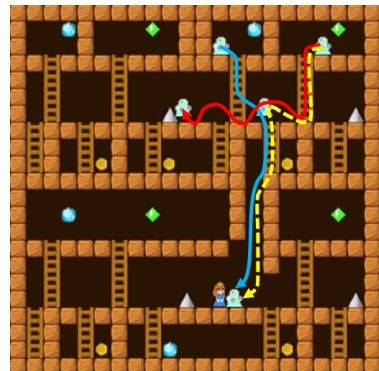

Figure 6: Visualization of trajectories. The blue line and red line visualize two policies using episodic memory, while the yellow dash line represents the combinatorial trajectory by value propagation in associative memory.

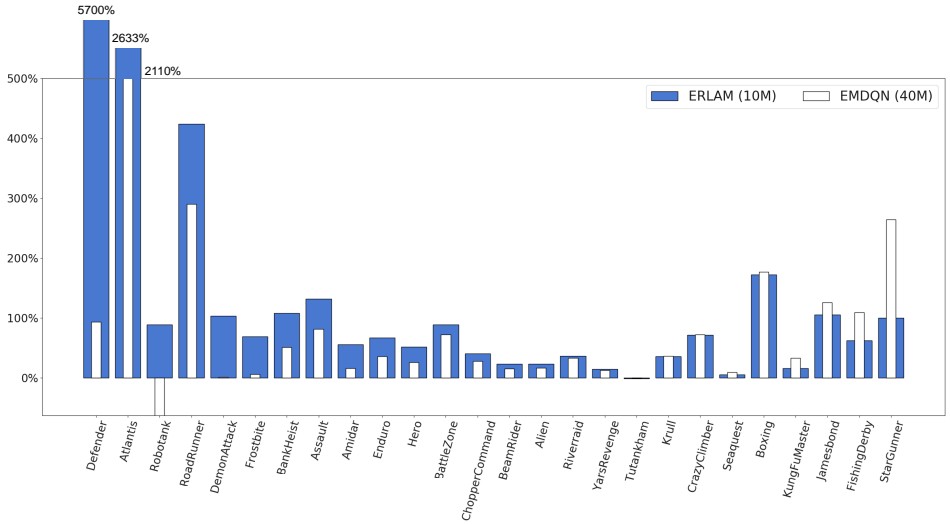

Figure 7: Comparison between ERLAM (i.e., DQN with associative memory) and EMDQN (i.e., DQN with episodic memory) measured in improvements of scores over DQN as shown in Eq. 6. Bars indicate how much each algorithm outperforms the DQN (i.e., DQN with no memory) agent. 0% means the performance is equal to DQN and higher is better.

Table 1: Performance comparisons on mean and median human-normalized scores as in (Mnih et al., 2015). All agents are trained using 10 million frames except for EMDQN which is trained with 40 million frames.

|        | DQN  | A3C  | Prior. DQN | MFEC | NEC   | EVA   | EMDQN(40M) | ERLAM |
|--------|------|------|------------|------|-------|-------|------------|-------|
| Mean   | 83.6 | 40.1 | 116.6      | 77.7 | 106.1 | 172.2 | 250.6      | **515.4** |
| Median | 16.0 | 6.9  | 32.3       | 40.9 | 53.3  | 39.2  | 95.5       | **103.5** |

We tested ERLAM on 25 popular and challenging Atari games. To evaluate our approach, we follow Wang et al. (2015) and measure improvement in percentage in score over the better of human and DQN agent scores for both ERLAM and EMDQN:

$$\frac{\text{Score}_{\text{Agent}} - \text{Score}_{\text{DQN}}}{\max\{\text{Score}_{\text{Human}}, \text{Score}_{\text{DQN}}\} - \text{Score}_{\text{Random}}}. \tag{6}$$

To test the sample efficiency of our method, we limit our training data to 10 million frames and compare with state-of-the-art results on episodic RL (i.e., EMDQN (Lin et al., 2018)), which are trained with 40 million frames and reported in their original paper. The results are shown in Figure 7. We found that even though our agent uses 4 times fewer training samples than EMDQN, ERLAM still outperforms EMDQN on 17 games. Overall, ERLAM significantly outperforms all baselines over most games. This suggests that associative memory can efficiently guide the learning of a parametric RL agent, and our framework of combining associative memory with parametric RL can achieve significantly better sample efficiency than existing RL algorithms. For the games where ERLAM does not perform very well, we summarize the reasons as follows. First, ERLAM is good at improving the sample-efficiency in near-deterministic environments but may suffer from overestimation in highly stochastic environments, such as Tutankham. Second, since representations learning is not the focus of this paper, we simply use the naive random projection as the state representations in memory. Random projection is only used for dimension reduction and does not contain useful high-level features or knowledge (e.g., objects and relations). Thus in some games with rarely revisited states, there are not enough joint nodes in our graph, and our algorithm does not perform well, such as FishingDerby and Jamesbond. In addition, We compare the overall performance (mean and median) of ERLAM with other methods in Table 1, which also shows that ERLAM has the best performance.

To gain a better understanding of our superior performance, we further plot learning curves (Figure 8) on four games, which include three general good cases (*Atlantis*, *BattleZone*, *StarGunner*) and a bad case (*BankHeist*) to demonstrate when associative memory works extremely well and when it is

not particularly effective. In addition, we plot the average values of states in memory (Figure 8) for better revealing the performance difference on game scores. Across most games, ERLAM is significantly faster at learning than EMDQN and DQN, but ERLAM only has a slightly better performance than EMDQN on *BankHeist*. The reasons lie in two folds. Firstly, there are more crossed experiences on *Atlantis*, *BattleZone*, *StarGunner* than *BankHeist*. Thus on the first three games, the values computed by associative memory are significantly larger than those in episodic memory. Secondly, we observe that the background objects in *BankHeist* have abnormally changeable appearance and complex behaviors, which are intractable for memory-based methods (e.g., MFEC, NEC, EMDQN, and ERLAM), especially with a simple random projection embedding function for state feature abstraction (we also discuss this in Conclusion Section). It also accounts for the reason why ERLAM and EMDQN have similar performance with DQN on this game.

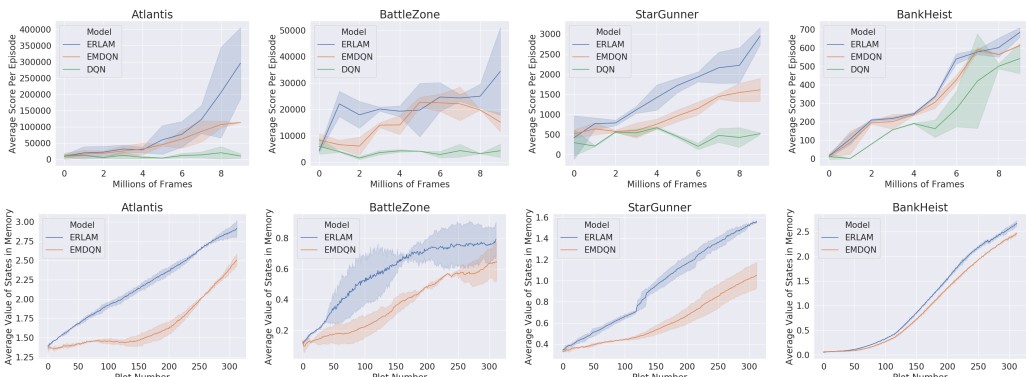

Figure 8: Examples of learning curves on 10 million frames compared with EMDQN and DQN. The top row shows average scores per episode and the bottom row shows average values of states in memory. One Plot Number is equivalent to about 30K frames. Note that 0 indicates the first million.

We also add experiments to verify our superior performance benefits from associative memory rather than representations (e.g., random projection). As shown in Appendix Figure 9, DQN with only random projections as inputs has much worse performance than ERLAM and the vanilla DQN, which suggests that it is associative memory that matters.

## 6 CONCLUSION

In this paper, we propose a biologically inspired sample efficient reinforcement learning framework, called *Episodic Reinforcement Learning with Associative Memory* (ERLAM). Our method explicitly organizes memorized states as a graph. We develop an efficient reverse-trajectory propagation strategy to allow the values of new experiences to propagate to all memory items through the graph rapidly. Experiments in the navigation domain and Atari games demonstrate that our proposed framework can significantly improve the sample efficiency of current reinforcement learning algorithms.

In the future, there are some interesting research directions that can be pursued within our proposed framework. Firstly, in this paper, following the work of Blundell et al. (2016) and Lin et al. (2018), our state embedding function $\phi$ is implemented as random projection. It is possible to incorporate advanced representation learning approaches that can capture useful features into our framework to support more efficient memory retrieval and further boost up performance. Secondly, existing episodic reinforcement learning algorithms mainly focus on value-based methods. It will be an interesting future work to extend episodic memory to policy gradient methods. Thirdly, we instantiate our associative memory in the learning phase in this paper. However, associative memory can also be used in explicit episodic control to enhance exploitation further. Fourthly, at the current stage, ERLAM, as a kind of episodic RL approach, is only good at improving sample-efficiency in near-deterministic environments. To deal with completely stochastic environments, our model can be potentially extended by storing the distribution of Q values (Bellemare et al., 2017; Dabney et al., 2018) instead of the maximum Q value in the associative memory.

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

# A    THEORETICAL CONVERGENCE

*Proof.* Note that our graph-based value propagation is similar to the proof of value iteration (Bellman, 1966; Bertsekas et al., 1995; Sutton & Barto, 2011). For any estimate of our graph-based action-value function $\hat{Q}_{\mathcal{G}}$

$$B\hat{Q}_{\mathcal{G}}(s,a) = R(s,a) + \gamma \max_{a' \in \mathcal{A}} \sum_{s' \in \mathcal{S}} P_{\mathcal{G}}(s'|s,a)\hat{Q}_{\mathcal{G}}(s',a'),$$

where $P_{\mathcal{G}}(s'|s,a)$ defines transition probability given graph $\mathcal{G}$. For any action-value function estimates $\hat{Q}_{\mathcal{G}}^1, \hat{Q}_{\mathcal{G}}^2$,

$$|B\hat{Q}_{\mathcal{G}}^1(s,a) - B\hat{Q}_{\mathcal{G}}^2(s,a)|$$

$$=\gamma \left| \max_{a' \in \mathcal{A}} \sum_{s' \in \mathcal{S}} P_{\mathcal{G}}(s'|s,a)\hat{Q}_{\mathcal{G}}^1(s',a') - \max_{a' \in \mathcal{A}} \sum_{s' \in \mathcal{S}} P_{\mathcal{G}}(s'|s,a)\hat{Q}_{\mathcal{G}}^2(s',a') \right|$$

$$\leq\gamma \max_{a' \in \mathcal{A}} \left| \sum_{s' \in \mathcal{S}} P_{\mathcal{G}}(s'|s,a)\hat{Q}_{\mathcal{G}}^1(s',a') - \sum_{s' \in \mathcal{S}} P_{\mathcal{G}}(s'|s,a)\hat{Q}_{\mathcal{G}}^2(s',a') \right|$$

$$=\gamma \max_{a' \in \mathcal{A}} \sum_{s' \in \mathcal{S}} P_{\mathcal{G}}(s'|s,a)|\hat{Q}_{\mathcal{G}}^1(s',a') - \hat{Q}_{\mathcal{G}}^2(s',a')|$$

$$\leq\gamma \max_{s \in \mathcal{S}, a \in \mathcal{A}} |\hat{Q}_{\mathcal{G}}^1(s,a) - \hat{Q}_{\mathcal{G}}^2(s,a)|$$

So the contraction property of Bellman operator holds:

$$\max_{s \in \mathcal{S}, a \in \mathcal{A}} |B\hat{Q}_{\mathcal{G}}^1(s,a) - B\hat{Q}_{\mathcal{G}}^2(s,a)| \leq \gamma \max_{s \in \mathcal{S}, a \in \mathcal{A}} |\hat{Q}_{\mathcal{G}}^1(s,a) - \hat{Q}_{\mathcal{G}}^2(s,a)| \tag{7}$$

For the fixed point $Q_{\mathcal{G}}^*$, we have:

$$\max_{s \in \mathcal{S}, a \in \mathcal{A}} |B\hat{Q}_{\mathcal{G}}(s,a) - B\hat{Q}_{\mathcal{G}}^*(s,a)| \leq \gamma \max_{s \in \mathcal{S}, a \in \mathcal{A}} |\hat{Q}_{\mathcal{G}}(s,a) - \hat{Q}_{\mathcal{G}}^*(s,a)| \implies \hat{Q}_{\mathcal{G}} \to Q_{\mathcal{G}}^* \tag{8}$$

Therefore, we prove that our graph-based value propagation algorithm can converge to unique optimal value.

$\square$

## B  RAW SCORES ON ATARI GAMES

Table 2: Raw scores on Atari games at 10 million frames. All agents are trained using 10 million frames except for EMDQN which is trained with 40 million frames.

| | DQN | A3C | Prior.DQN | MFEC | NEC | EVA | EMDQN(40M) | ERLAM |
|---|---|---|---|---|---|---|---|---|
| Alien | 634.80 | 415.50 | 800.50 | 1717.70 | **3460.60** | 1007.93 | 1662.00 | 2070.85 |
| Amidar | 126.80 | 96.30 | 99.10 | 370.90 | 811.30 | 231.19 | 374.10 | **980.47** |
| Assault | 1489.50 | 720.80 | 1339.90 | 510.20 | 599.90 | 550.77 | 2566.80 | **3230.18** |
| Atlantis | 14210.50 | 36383.00 | 12579.10 | 95499.40 | 51208.00 | 180367.20 | 290953.30 | **359530.00** |
| BankHeist | 29.30 | 15.80 | 70.10 | 163.70 | 343.30 | **4022.45** | 348.00 | 702.92 |
| BattleZone | 6961.00 | 2354.20 | 13500.00 | 19053.60 | 13345.50 | 14000.47 | 28300.00 | **33095.24** |
| BeamRider | 3741.70 | 450.20 | 3249.60 | 858.80 | 749.60 | 1914.30 | 5980.90 | **7116.67** |
| Boxing | 31.30 | 2.50 | 64.70 | 10.70 | 72.80 | 58.43 | **89.30** | 87.77 |
| ChopperCommand | 827.20 | 1036.70 | 1426.50 | 3075.60 | **5070.30** | 1612.93 | 3106.70 | 4172.83 |
| CrazyClimber | 66061.60 | 70103.50 | 76574.10 | 9892.20 | 34344.00 | 90656.27 | **107038.70** | 106538.71 |
| Defender | 2877.90 | 4596.00 | 3486.40 | 10052.80 | 6126.10 | 2890.44 | 14408.00 | **705833.33** |
| DemonAttack | 5541.90 | 346.80 | 6503.60 | 1081.80 | 641.40 | 504.52 | 5603.10 | **11056.75** |
| Enduro | 364.90 | 0.00 | **1125.80** | 0.00 | 1.40 | 1106.35 | 659.00 | 912.73 |
| FishingDerby | -81.60 | -89.50 | -48.20 | -90.30 | -72.20 | -68.10 | **8.40** | -30.20 |
| Frostbite | 339.10 | 218.90 | 711.30 | 925.10 | 2747.40 | 1005.44 | 596.30 | **3193.83** |
| Hero | 1050.70 | 4598.20 | 5164.50 | 14767.70 | **16265.30** | 12075.89 | 7247.80 | 13615.00 |
| Jamesbond | 165.90 | 31.50 | 203.80 | 244.70 | 376.80 | 252.18 | **586.70** | 518.42 |
| Krull | 6015.10 | 3627.60 | 6700.70 | 4555.20 | 5179.20 | 4030.04 | **7798.30** | 7755.80 |
| KungFuMaster | 17166.10 | 6634.60 | 21456.20 | 12906.50 | **30568.10** | 25005.15 | 23890.00 | 20353.49 |
| Riverraid | 3144.90 | 2312.60 | 4871.80 | 4195.00 | 5498.10 | 4026.74 | 7728.30 | **8138.66** |
| RoadRunner | 7285.40 | 759.90 | 24746.60 | 5432.10 | 12661.40 | 28194.17 | 27856.60 | **37318.63** |
| Robotank | 14.60 | 2.40 | 8.50 | 7.30 | 11.10 | 15.13 | 5.30 | **25.44** |
| Seaquest | 618.70 | 514.10 | 1192.20 | 711.60 | 1015.30 | 1714.15 | **4235.90** | 2693.96 |
| StarGunner | 604.80 | 613.60 | 1131.40 | 14843.90 | 1171.40 | 2006.22 | **23933.30** | 9432.97 |
| Tutankham | 148.70 | 108.30 | **194.00** | 86.30 | 121.60 | 171.33 | 148.00 | 146.94 |
| YarsRevenge | 7614.10 | 9953.00 | 9228.50 | 5956.70 | **21490.50** | 11010.90 | 13236.70 | 14259.26 |

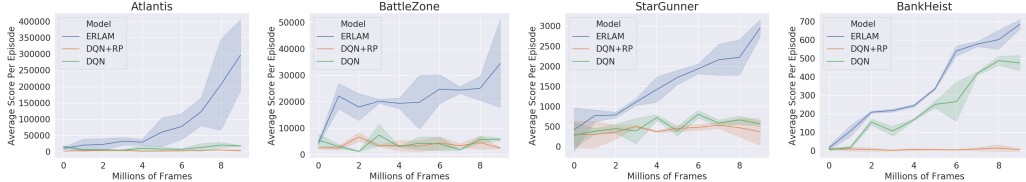

Figure 9: Learning curves on 10 million frames compared with ERLAM, DQN with random project and vanilla DQN.

