# OpenReview forum: "Episodic Reinforcement Learning with Associative Memory"
_ICLR.cc/2020/Conference — Accept (Poster)_

### Official Review · AnonReviewer3 · 2019-10-23
**Official Blind Review #3**

**Rating:** 6

**Review:**

The paper proposes to combine DQN with a nonparametric estimate of the optimal Q function based on the graph of all observed transitions in the buffer. Specifically, they use the nonparametric estimate as a regularizer in the DQN loss. They show that this regularizer facilitates learning, and compare to other nonparametric approaches. I found the paper easy to read. The ideas are intuitive and seem to work.

It would be great to have more experiments providing insight into when the associative memory estimate works and when it doesn't. Since at the end of the day both DQN and the non-parametric estimate use the same data, there's no fundamental reason why the later should contain more information. Is it possible that more aggressive training of DQN would eliminate the need for the nonparametric estimate? Why would I expect the nonparametric estimate based on random projections to generalize better to new states than DQN? What would be the performance of DQN with only the random projections as inputs? I believe including experiments probing in this direction would make the paper better.

-------------------------------------------------------------------------------------------------------------------
Thanks for your response and the additional experiments. I still find the paper interesting and hence keeping my score as is.

**Experience Assessment:**

I have read many papers in this area.

**Review Assessment: Checking Correctness Of Derivations And Theory:**

N/A

**Review Assessment: Checking Correctness Of Experiments:**

I assessed the sensibility of the experiments.

**Review Assessment: Thoroughness In Paper Reading:**

I made a quick assessment of this paper.

---

> ### Author Response · Authors · 2019-11-14
> **Response**
>
> Thank you for the thoughtful comments and suggestions.
>
> DQN and the non-parametric estimation seems to use the same data and learn based on same information eventually. However, DQN distills the information much slower than the non-parametric estimation because of the following three reasons.
>
> (1) Since DQN uses neural networks with stochastic gradient descent (SGD) optimization, small learning rates are required to stabilize training during the optimization on random minibatchs, and high learning rates can cause catastrophic interference due to the global approximation nature of neural networks [1]. However, an agent with a non-parametric memory can directly latch on successful policies as soon as they are experienced instead of waiting for many steps of optimization. The requirement for slow updating in SGD also accounts for why more aggressive training of DQN does not help. Factually, we have tried many aggressive settings of hyperparameters for DQN, and found that the hyperparameters reported in the original DQN paper are the best (which are also used in our paper as baseline).
>
> (2) DQN randomly samples experience tuples from replay buffer to update value function, which neglects the trajectory nature of an agent’s experience (i.e., one tuple occurs after another and so information of the next state should be quickly propagated into the current state). Instead, we build a graph based on transitions of states and associate related experience trajectories, which enables both intra-episode and inter-episode value propagation (the inter-episode path is like augmentation of counterfactual combinatorial trajectories). When a reward signal is first discovered at one state, the related states can quickly receive this information through the intra-episode or inter-episode non-parametric value propagation.
>
> (3) DQN uses random sampling for value-bootstrapping, while we develop an efficient reverse-trajectory propagation strategy to allow rapid value propagation through the graph. Recently, an intra-episode value-bootstrapping has been adopted in RL [2] and demonstrates superior performance of reverse-trajectory propagation compared to random sampling.
>
> We also add experiments to verify our superior performance benefits from associative memory rather than representations (e.g., random projection). As shown in Appendix Figure 5, DQN with only random projections as inputs has a much worse performance than the vanilla DQN. Because random projection is a very simple representation that is only used for dimension reduction and does not contain useful high-level feature or knowledge (e.g., objects and relations). Our future work will focus on incorporating advanced representation learning approaches that can capture useful features into our framework to support more efficient memory retrieval and further boost up the performance.
>
> We have added these results and descriptions in the revised paper. We would like to thank you again for the thoughtful questions and constructive suggestions.
>
>
> References:
> [1] McCloskey, M., & Cohen, N. J. (1989). Catastrophic interference in connectionist networks: The sequential learning problem. In Psychology of learning and motivation (Vol. 24, pp. 109-165). Academic Press.
> [2] Lee, S. Y., Choi, S., & Chung, S. Y. (2018). Sample-efficient deep reinforcement learning via episodic backward update. arXiv preprint arXiv:1805.12375.

---

### Official Review · AnonReviewer2 · 2019-10-23
**Official Blind Review #2**

**Rating:** 3

**Review:**

The paper proposes a new method for organizing episodic memory in with deep Q-networks. It organizes the memory as a graph in which nodes are internal representations of observed states and edges link state transitions. Additionally, nodes from different episodes are merged into a single node if they represent the same state, allowing for inter-episode value propagation of stored rewards. The authors experimentally evaluate the method against reasonable baselines and show improved performance in the majority of tasks.

While the proposed method seemingly leads to better performance, the analysis of the results appears superficial. First, it lacks theoretical rigor to explain why the proposed propagation mechanism works, for instance, a proof of the optimal substructure in Eq. (3) would be helpful. While I see the authors mentioned previous work which do not have optimal substructures and mention this for the case of navigation-like tasks, the matter is not discussed and unclear in other scenarios.

This in itself would not be much of an issue if the experiments highlighted advantages and limitations of the proposed method, but that is not the case. For instance, per-task comparisons to EVA (mentioned in the paper) could indicate how useful inter-episode value propagation was in each task.  Table 1 mentions EVA, but only on aggregated results, thus not providing insight into this. Furthermore, the experiments selected to be detailed in Figure 4 are, in my opinion, suboptimal choices, as the worst-performing of the selected tasks is still better than the baselines. It would be more insightful to also compare those whose performance is close to the baselines, such as the Boxing environment, and significantly worse than baselines, such as the StarGunner environment (as shown in Figure 3). With those issues in mind, my conclusion is to recommend to reject the paper in the current state.

Minor comments:
- Thoroughly review the writing and grammar, the text in its current state needs significant improvement in this regard
- Equation references missing parentheses
- Introduction, 1st paragraph, second sentence is incomplete
- Algorithm 2: tuples taken from/appended to sets G and D are not consistent (cf. lines 11 and 13)
- Figure 3: The large amount of bars in the plot would benefit from horizontal lines across the plot for each tick on the y axis
- Table 1: caption before the table; should also state the meaning of numbers (percentage or normalized score, calculated from what?)
- Figure 4: labels (all text except plot titles) are impossible to read in print
- Section 3 could be placed before Section 2, laying the mathematical framework, and then following the discussion with related work
- There is redundant content in Sections 2 and 3


----

I am happy with the authors response and changed the score accordingly

**Experience Assessment:**

I have read many papers in this area.

**Review Assessment: Checking Correctness Of Derivations And Theory:**

I assessed the sensibility of the derivations and theory.

**Review Assessment: Checking Correctness Of Experiments:**

I carefully checked the experiments.

**Review Assessment: Thoroughness In Paper Reading:**

I read the paper at least twice and used my best judgement in assessing the paper.

---

> ### Author Response · Authors · 2019-11-14
> **Response**
>
> Thank you for the thoughtful comments and suggestions. In the following, we address the concerns point by point.
>
> Q1. First, it lacks theoretical rigor to explain why the proposed propagation mechanism works.
>
> A1. We add a theoretical proof in Appendix A that our graph-based value propagation algorithm can converge to unique optimal value in the general RL setting not only the navigation tasks.
>
> Q2. This in itself would not be much of an issue if the experiments highlighted advantages and limitations of the proposed method, but that is not the case.
>
> A2. We would like to thank you again for the constructive suggestions. We have added more experimental details as your suggested. We add per-tasks comparisons to baseline models (i.e., DQN, A3C, Prioritized DQN, MFEC, NEC, EVA, EMDQN, and ERLAM) in Appendix Table 2. We stress that ERLAM does not perform significantly worse than baselines on StarGunner. Figure 3 shows scores of EMDQN with 40M samples while ERLAM uses only 10M samples. As shown in Figure 4, ERLAM significantly outperforms EMDQN on StarGunner when they both use 10M samples. As mentioned in A1 to Reviewer #1, our algorithm is good at improving the sample-efficiency in near-deterministic environments but may suffer from overestimation in highly stochastic environments, such as Tutankham.  In addition, since representations learning is not the focus of this paper, we simply use the naive random projection as the state representations in memory. As discussed in response to Reviewer #3, random projection is only used for dimension reduction and does not contain useful high-level feature or knowledge (e.g., objects and relations). In some games with rare revisited states, there are not enough joint nodes in our graph and thus our algorithm does not perform well, such as FishingDerby and Jamesbond.
>
> Q3. Minor comments.
>
> A3. We have refined all the language you mentioned and improved the presentation in the revised paper as you suggested.

---

### Official Review · AnonReviewer1 · 2019-10-26
**Official Blind Review #1**

**Rating:** 6

**Review:**

This paper proposes Episode Reinforcement Learning with Associative Memory (ERLAM), which maintains a graph based on the state transitions (i.e. nodes correspond to states, and edges correspond to transitions) and propagates the values through the edges in the graph in the reverse order of each trajectory. The learned associative memory is then used for the regularization loss for training Q-network. Experimental results show that ERLAM significantly improves the sample efficiency in Atari benchmarks.

- Overall, the paper is well-motivated and easy to follow. The experimental results demonstrate that the proposed method is promising. For the states that are already visited, instead of simply replacing the value to the better return (i.e. Eq. (1)), ERLAM makes join points to connect different trajectories, which enables further improvement via Bellman optimality-like backup (i.e. Eq. (3)).

- Can ERLAM deal with a stochastic environment? It seems that ERLAM would more likely to over-estimate the values for the state than the existing episodic RL algorithms.

- In order to make join points, it should be possible to determine whether two features of states are equal or not. How was this determined? It seems rare to reach the 'exact' same feature in Atari domains (i.e. 4 consecutive frames should be equal).

- In Algorithm 2, the pseudo-code is somewhat confusing in that R_t is appended to G before the episode ends.

**Experience Assessment:**

I have read many papers in this area.

**Review Assessment: Checking Correctness Of Derivations And Theory:**

N/A

**Review Assessment: Checking Correctness Of Experiments:**

I assessed the sensibility of the experiments.

**Review Assessment: Thoroughness In Paper Reading:**

I read the paper at least twice and used my best judgement in assessing the paper.

---

> ### Author Response · Authors · 2019-11-14
> **Response**
>
> Thank you for your thoughtful comments and constructive suggestions. In the following, we address the concerns point by point.
>
> Q1. Can ERLAM deal with a stochastic environment? It seems that ERLAM would more likely to over-estimate the values for the state than the existing episodic RL algorithms.
>
> A1. Episodic control mainly focuses on improving the sample-efficiency in near-deterministic environments [1,2,3]. In stochastic environments, both previous episodic RL algorithms and ERLAM may suffer from over-estimation during the maximization update of revisited states in memory. However, if one state-action pair has many different subsequent states, we will select the recently sampled one instead of the maximum to build the graph, which amounts to using the expectation of next states from a long term view and contributes to alleviate the over-estimation problem. Thus, ERLAM can work well in near-deterministic environments and our experiments also confirm this. In our experiments, although there are about 34% stochastic states in each game on average, ERLAM significantly outperforms the baselines, which suggests our algorithm is able to handle environments with some randomness. In addition, for a completely stochastic environment, our model can be extended by storing distribution of Q values [4,5] instead of the maximum Q value in the associative memory, which is an important direction for future work. We have added these descriptions in our revised paper and thank you again for the insightful question.
>
> Q2. In order to make join points, it should be possible to determine whether two features of states are equal or not. How was this determined? It seems rare to reach the 'exact' same feature in Atari domains (i.e. 4 consecutive frames should be equal).
>
> A2. We set a small threshold (i.e., 0.0000001) to determine whether two features of states are equal or not. We use the same threshold in all games. As reported in [1], Atari games have a sizeable percentage of states-action pairs that are exactly the same. For example, there are about 10% repeating states in Frostbite, 60% in Q*bert, 50% in Ms. PAC-MAN, 45% in Space Invaders, and 10% in River Raid. Actually, some states are not exactly the same but extremely similar (below our threshold) so they can also form joint points in our graph. In our experiments, a great number of joint states (about 1k on average) will be updated during each running of our value propagation algorithm.
>
> Q3. In Algorithm 2, the pseudo-code is somewhat confusing in that R_t is appended to G before the episode ends.
>
> A3. R_t is appended to G computed after each episode ends. We have modified Algorithm 2 in the revised paper as you suggested.
>
> References:
> [1] Blundell, C., Uria, B., Pritzel, A., Li, Y., Ruderman, A., Leibo, J. Z., ... & Hassabis, D. (2016). Model-free episodic control. arXiv preprint arXiv:1606.04460.
> [2] Pritzel, A., Uria, B., Srinivasan, S., Badia, A. P., Vinyals, O., Hassabis, D., ... & Blundell, C. (2017, August). Neural episodic control. In Proceedings of the 34th International Conference on Machine Learning-Volume 70 (pp. 2827-2836). JMLR. org.
> [3] Gershman, S. J., & Daw, N. D. (2017). Reinforcement learning and episodic memory in humans and animals: an integrative framework. Annual review of psychology, 68, 101-128.
> [4] Marc G Bellemare, Will Dabney, and Rémi Munos. (2017). A distributional perspective on reinforcement learning. In Proceedings of the 34th International Conference on Machine Learning-Volume 70, pages 449–458. JMLR. org.
> [5] Will Dabney, Georg Ostrovski, David Silver, and Remi Munos. (2018) Implicit quantile networks for distributional reinforcement learning. In International Conference on Machine Learning, pages 1104–1113.

---

### Author Response · Authors · 2019-11-15
**Overall response**

We thank all reviewers for their efforts and thoughtful comments, which are helpful for improving the quality of our paper. In the updated paper, we have revised our manuscript according to their suggestions. Below, we describe in detail how we have modified our paper to address the reviewers’ feedback.

1. We add a theoretical proof (Appendix A) to prove that our graph-based value propagation algorithm can converge to unique optimal value. Our algorithm provides a non-parametric estimation of the optimal Q function based on the graph of all observed transitions. This non-parametric estimation can be used as a lower bound for the parametric Q network and thus help boost up the learning of Q network.

2. We clarify our contributions focus on near-deterministic environments that are always taken as a basic assumption in conventional episodic reinforcement learning. We also discuss how stochastic environments affect our algorithm and provide possible ways to extend it to highly stochastic environments.

3. We add experiments to verify our superior performance benefits from associative memory rather than representations (e.g., random projection).

4. We refine the presentation of our paper as suggested by the reviewers, and add additional descriptions to discuss when our method works and when it doesn't.

---

### Decision · Program_Chairs · 2019-12-19

**Decision:**

Accept (Poster)

**Comment:**

The submission tackles the problem of data efficiency in RL by building a graph on top of the replay memory and propagate values based on this representation of states and transitions. The method is evaluated on Atari games and is shown to outperform other episodic RL methods.

The reviews were mixed initially but have been brought up by the revisions to the paper and the authors' rebuttal. In particular, there was a concern about theoretical support and the authors added a proof of convergence. They have also added additional experiments and explanations. Given the positive reviews and discussion, the recommendation is to accept this paper.